# The Effectiveness and Safety of Wu Tou Decoction on Rheumatoid Arthritis—A Systematic Review and Meta-Analysis

**DOI:** 10.3390/healthcare12171739

**Published:** 2024-08-31

**Authors:** Jeong-Hyun Moon, Gyoungeun Park, Chan-Young Kwon, Joo-Hee Kim, Eun-Jung Kim, Byung-Kwan Seo, Seung-Deok Lee, Seung-Ug Hong, Won-Suk Sung

**Affiliations:** 1College of Korean Medicine, Dongguk University Graduate School, Seoul 04620, Republic of Korea; 2015213276@dgu.ac.kr (J.-H.M.); 2013120530@dgu.ac.kr (G.P.); 2Department of Oriental Neuropsychiatry, College of Korean Medicine, Dongeui University, Busan 47340, Republic of Korea; beanalogue@deu.ac.kr; 3Department of Acupuncture and Moxibustion Medicine, College of Korean Medicine, Sangji University, Wonju-si 26339, Republic of Korea; jhkim712@sangji.ac.kr; 4Department of Acupuncture & Moxibustion, Dongguk University Bundang Oriental Hospital, Seongnam-si 13601, Republic of Korea; hanijjung@dumc.or.kr; 5Department of Acupuncture and Moxibustion Medicine, Kyung Hee University College of Korean Medicine, Kyung Hee University Hospital at Gangdong, Seoul 02447, Republic of Korea; sbkacu@khu.ac.kr; 6Department of Acupuncture & Moxibustion, Dongguk University Ilsan Oriental Hospital, Goyang-si 10326, Republic of Korea; chuckman@dongguk.edu; 7Department of Ophthalmology, Otolaryngology and Dermatology, Dongguk University Ilsan Oriental Hospital, Goyang-si 10326, Republic of Korea; heenthsu@duih.org

**Keywords:** rheumatoid arthritis, Wu tou decoction, conventional treatment, systematic review, meta-analysis

## Abstract

Rheumatoid arthritis (RA) is an autoimmune disease primarily affecting the joints and requires various treatments, including medication, injection, and physiotherapy. Wu tou decoction (WTD) is a traditional Chinese medicine prescribed for RA, with several articles documenting its effectiveness in RA treatment. This systematic review and meta-analysis aimed to evaluate the efficacy and safety of WTD for RA. We searched for randomized controlled trials (RCTs) comparing WTD with conventional treatments (including medication, injection, and physiotherapy) from its inception to May 2024. Primary outcomes were disease activity scores, including effective rate, tender joint count, and morning stiffness. Secondary outcomes comprised blood test results (erythrocyte sedimentation rate, C-reactive protein, and rheumatoid factor) and adverse events. Nineteen RCTs involving 1794 patients were included. Statistically, WTD demonstrated better improvement than conventional treatments (18 medications and 1 injection) across the effective rate, joint scale, and blood tests, regardless of the treatment type (monotherapy or combination therapy). Adverse events were reported in 11 studies, with no statistical differences observed between them. The numerical results showed that WTD may offer potential benefits for managing RA. However, the significant discrepancy between clinical practice and the low quality of the RCTs remains a limitation. Therefore, further well-designed studies with larger patient cohorts are needed to draw definitive conclusions.

## 1. Introduction

Rheumatoid arthritis (RA) is a chronic autoimmune disease characterized by persistent inflammation in the joints and other organs [1], leading to immune system dysfunction [2]. RA can develop at any age and affects 0.5–1% of the global population [3,4]. The etiology is unknown, but tumor necrosis factor (TNF)-α and interleukin (IL)-6 are reported to play important roles in the pathogenesis and maintenance of inflammation in RA [5]. Additionally, T cells play an important role in bone destruction and inflammation, and B cells are the main source of production of autoantibodies, such as rheumatoid factor (RF), anti-citrullinated peptide antibody (ACPA), cytokine secretion, and antigen presentation [6,7]. In the later stages of the disease, macrophages produce cytokines, while dendritic cells and natural killer cells also fulfill their duties [8].

RA is characterized by arthritis that can cause joint damage, systemic inflammation, and extraarticular symptoms in other organs, including the heart, kidney, lungs, digestive system, eye, skin, and nervous system [9]. In extra-articular symptoms, approximately 30% of patients present with rheumatoid nodules, whereas 10% of patients suffer from Sjogren’s syndrome [10], and excessive complications lead to increased mortality [9]. Its diagnosis involves abnormal erythrocyte sedimentation rate (ESR), C-reactive protein (CRP), and RF levels, in addition to joint swelling, autoantibody production, and duration of symptoms [11]. ACPA is also specific to RA. Like CRP elevation, the presence of ACPA and RF indicates a very early onset of RA development [12].

In clinical practice, several treatment strategies and conventional treatments for RA are available [13,14], with the most common being medications, including nonsteroidal anti-inflammatory drugs (NSAIDs) [15], glucocorticoids, and disease-modifying antirheumatic drugs (DMARDs) [16]. Recent recommendations are to start DMARDs plus glucocorticoids immediately. Effective doses of methotrexate (MTX, oral or subcutaneous) are used for 3–6 months [17]. If medication is ineffective in controlling RA symptoms, treatment is rapidly expanded to include various medications with a treat-to-target strategy [18]. In addition, various treatments are administered to alleviate symptoms and enable daily activities with regular disease activity monitoring [18]. In contrast, the use of natural ingredients from Oriental medicine offers a viable alternative for RA treatment. Consequently, healthcare providers and patients are increasingly interested in traditional Chinese medicine (TCM), particularly Wu tou decoction (WTD) [19].

WTD is an herbal medicine that has been used to treat joint-related diseases and consists of *Radix Aconiti* (Wu tou), *Herba Ephedrae* (Ma huang), *Radix Astragali* (Huang qi), *Radix Paeoniae Alba* (Bai shao), and *Radix Glycytthiza* (Gan cao) [20]. There have been attempts to determine the pharmacological component, action, and mechanism of WTD. Chemical profiling of WTD in the rat model via high-performance liquid chromatography revealed that *Radix Aconiti* and *Herba Ephedrae* contained alkaloids that had anti-inflammatory and analgesic effects, *Radix Astragali* and *Radix Glycytthiza* exert antioxidant effects on flavones and glucosides, while monoterpene glycosides from *Radix Paeoniae alba* have neuroprotective effects [21,22].

Regarding the active components, benzoylaconitine, benzoylneoaconitine, benzoylhypoaconitine (from Wu tou), ephedrine hydrochloride, pseudoephedrine hydrochloride (from Ma huang), paeoniflorin (from Bai shao), verbasil-7-O-glucoside (from Huang qi), glycyrrhizinate, and glycyrrhizin (from Gan cao) were identified in the rat model [23]. Studies using an animal model of arthritis reported that WTD modulates C-C chemokine receptor 5 (CCR5), affects the inflammatory response in macrophages [24], inhibits nuclear factor kappa B (NF-κB) phosphorylation (through the action of *Herba ephedrae*), and enhances nuclear factor-like 2 (Nrf2) expression (via *Radix Astragali* and *Radix aconiti*) [25]. Other mechanisms include reducing angiogenesis in the joint synovium by inhibiting VEGF165/MH7A, which is crucial for endothelial cell activation [26].

These active components, receptors, and mechanisms are closely related to RA. Benzoylaconitine inhibits the expression of IL-6 and IL-8 by inhibiting the activation of the mitogen-activated protein kinase (MAPK), Akt, and NF-κB pathways in human synovial cells [27]. In an arthritic rat model, pseudoephedrine reduces the expression of TNF-α, IL-β, and IL-6 while paeoniflorin additionally alters cyclooxugenase-2 protein expression [28,29]. CCR5 is a key gene that regulates the cellular immune response and cytokine signaling, which are crucial for distinguishing RA [30]. Nrf-2 regulates oxidative stress, immune response, and cartilage and bone metabolism [31]. Inflammation-related signals associated with NF-κB have been reported in the context of RA [32]. Thus, the possibility that WTD might alleviate symptoms of RA has increased [24,25,33].

Studies have examined the clinical efficacy of WTD in RA [34]; however, knowledge about its effects and safety remains limited due to the lack of a systematic review (SR). Therefore, this study aims to evaluate the clinical efficacy and adverse events of WTD in treating RA through an SR and meta-analysis.

## 2. Materials and Methods

### 2.1. Ethics

Ethical approval was not required because no personal information of patients was collected.

### 2.2. Study Registration

This SR followed the Preferred Reporting Items for Systematic Reviews and Meta-Analyses (PRISMA) Protocols 2020 statement [35]. The protocol was registered in PROSPERO (Registration number: CRD42022310337) and published in March 2022 [36].

### 2.3. Search Strategy

Data searches were conducted from inception to May 2024 across multiple databases, including MEDLINE, Cochrane Library, Web of Science, ScienceDirect, Wiley, EMASE, China National Knowledge Infrastructure, CiNii, Wanfang data, J-STAGE, KoreaMed, Korean Studies Information Service System, National Digital Science Library, Korea Institute of Science and Technology Information, and Oriental Medicine Advanced Searching Integrated System. Searches were performed in the appropriate language for each database (e.g., ‘Wu tou decoction’ and ‘rheumatoid arthritis’ in the English database). Additionally, related literature materials, reports, and papers were searched. Manual searches were also performed using textbooks on RA and by contacting authors via e-mail when necessary. 

### 2.4. Inclusion and Exclusion Criteria 

We set the following inclusion and exclusion criteria.

#### 2.4.1. Inclusion Criteria

We included studies involving patients with RA, regardless of age and sex. Randomized controlled trials (RCTs) were included, excluding those that omitted the “randomization” phase or implemented incorrect randomization. The study included research that used WTD as an experimental treatment for RA and compared its efficacy with conventional treatments, such as nonoperative methods, including medication, injection, and physiotherapy. 

#### 2.4.2. Exclusion Criteria

Patients with other forms of arthritis, such as osteoarthritis and gout, were excluded from the study. Non-RCTs, case reports, SR, and studies where the control group did not receive treatment or a placebo were excluded. Additionally, studies that newly initiated or changed interventions during the treatment period or did not clearly present pre- and post-experiment comparison values were excluded.

### 2.5. Study Selection and Data Extraction

Two reviewers (JHM and GEP) independently screened studies and extracted information. They excluded studies based on titles and abstracts and then reviewed the full texts of the articles included to assess their suitability for inclusion in this SR. Any disagreements were resolved through discussion or by involving an additional reviewer (WSS) responsible for reaching a final decision.

#### 2.5.1. The Characteristics of Study

The extracted information included first author, publication years, patient characteristics, interventions in the two groups, session frequencies, duration periods, outcome measures, results, adverse events, and quality of studies. In cases of incomplete information, attempts were made to contact the authors for complete data. If complete data were not obtainable, the meta-analysis was conducted using as much complete data as possible.

#### 2.5.2. Outcome Measures

According to the protocol [36], the disease activity score was established, with the effective rate (ER), tender joint count (TJC), and morning stiffness (MS) as primary outcome measures. Secondary outcome measures included blood test results such as ESR, CRP, RF, and adverse events. A meta-analysis was conducted if there were two or more studies that met the criteria for data synthesis.

### 2.6. Statistical Analysis

The differences from baseline to endpoint were combined to calculate the mean difference (MD) and 95% confidence intervals (CI) for the same outcome measures, while the standardized mean difference and 95% CI were calculated for different outcome measures. These were evaluated using either a random-effects or fixed-effects model. Review Manager (Version 5.3; Copenhagen; The Nordic Cochrane Center, The Cochrane Collaboration, 2014) software for Windows was utilized for the SR. Chi-squared and I-squared tests were employed to assess heterogeneity across the selected studies [37]. The interpretation of heterogeneity was categorized as follows: Heterogeneity levels of 0–40%, 30–60%, 50–90%, and 75–100% were classified as unimportant, moderate, substantial, and considerable, respectively. Subgroup analyses were performed when considered necessary.

### 2.7. Quality Assessment 

Two reviewers independently assessed the risk of bias using the “Risk of Bias” tool from the Cochrane Collaboration [38], which evaluates seven areas: sequence generation, allocation concealment, blinding of participants and investigators, blinding of outcome assessment, incomplete outcome data, selective reporting, and other biases. The risk of bias for each domain was categorized as “low risk”, “high risk”, or “unclear risk”. Disagreements between reviewers were resolved through discussions; if unresolved, an additional reviewer mediated the final decision. The quality of evidence was rated using the Grades of Recommendation, Assessment, Development, and Evaluation framework, starting from high-quality evidence and stepping down to moderate, low, and very low quality [39]. The quality of evidence was assessed for several outcomes. The primary outcomes included ER, TJC, and MS while secondary outcomes consisted of ESR, CRP, and RF. The quality of evidence was rated high if all included studies were RCTs. Five factors, namely, risk of bias, inconsistency, indirection, imprecision, and other considerations, could affect downgrading. Based on the seriousness of each factor after evaluation, a decision was made whether to downgrade by one or two grades. Considering the prevalence of RA, the optimal information size (OIS) was determined to be 200. Consequently, a sample size exceeding 200 was considered significant.

### 2.8. Publication Bias

If more than 10 studies are included in this SR, funnel plots will be presented.

## 3. Results

### 3.1. Study Selection

According to their protocol, articles were searched, resulting in the identification of 1677 studies in the databases. After removing 405 duplicate records, 1272 studies underwent screening based on their abstracts and titles. Overall, 1210 studies were excluded due to being non-RCT, non-RA, non-WTD, or for other reasons. After screening, 62 studies were retrieved and assessed for eligibility. The full text of these 62 studies was reviewed, resulting in the exclusion of 37 studies for the following reasons: (1) improper interventions, including modification of herbal medicine according to pattern identification (34 studies); (2) improper randomization (two studies); and (3) inability to obtain the full text (one study). Finally, 25 studies were included in this review, with 19 of them selected for synthesis in this SR and meta-analysis (Figure 1).

### 3.2. Characteristics of the Included Studies

Table 1 shows that our SR included 25 studies with 2189 participants. All studies were published in China between 2008 and 2022. In the 19 studies used for this meta-analysis, WTD was used in five studies as monotherapy for RA, including 426 participants [40,41,42,43,44], while WTD was used in the other 14 studies in combination with conventional treatment, including 1368 participants [45,46,47,48,49,50,51,52,53,54,55,56,57,58]. The effects of WTD were analyzed by separating the studies into monotherapy and combination therapy groups. In addition to the 19 included studies, six studies were considered but not included owing to insufficient data [34,59,60,61,62,63].

Regarding treatment, four studies [43,44,50,55] employed the original prescription of WTD, while the other 15 studies utilized a modified WTD. These 15 studies showed various options for modification. Gui zhi (*Cinnamomum cassia Presl*) [40,41,42,46,48,51,53,54,56] was the most frequently used herb for content modification in nine studies. Following closely were xi xin (*Asarum sieboldii*) [46,47,48,49,51,53] and du huo (*Aralia cordata*) [45,46,48,52,53], utilized in six and five studies, respectively. Finally, one study [47] used a fu fang wu tou microemulsion without specifying the contents of their modification. The longest treatment period was 6 months, while the shortest was 15 days. The number of WTD doses per day varied; however, twice a day was the most common, encompassing 12 studies [40,42,43,44,45,49,50,51,53,54,56,57]. 

In the control group, medication was used in 18 studies, while injection was [54] utilized in one (^99m^Tc-Methyl diphosphonate). Among the medications, the most used were DMARDs. MTX [42,43,44,45,46,48,51,55,57,58] and leflunomide (LEF) [40,41,44,46,49,50,52,55] were each employed in 10 and 8 studies, respectively. Lastly, NSAIDs were used in six studies [42,47,49,53,55,57], including nimesulide, meloxicam, diclofenac (DCF), and ibuprofen.

Regarding outcome measures, the ER was used most often (16 studies). TJC was used in five studies, while MS was used in four studies. Blood tests included ESR and CRP, each used in 10 studies, and RF, used in seven studies. Additionally, the disease activity score in 28 joints (DAS28) was used in six studies. Other RA-related scales utilized in the selected studies included ACR20, 50, 70, and the health assessment questionnaire (HAQ), each used in three studies.

### 3.3. Efficacy Assessment of WTD Monotherapy

WTD was used in five studies [40,41,42,43,44] as monotherapy (n = 426), with modified WTD used in three studies, while the original WTD was used in two. In the control group, MTX was used in two studies, while LEF was used in the other two. 

#### 3.3.1. Effective Rate

The pooled results showed a significant difference between the WTD and control groups (Risk ratio [RR] = 1.25, 95% CI: 1.14–1.37, *p* < 0.00001). The I^2^ value was 48%, indicating moderate heterogeneity among the three studies [40,41,42] (n = 338) (Figure 2).

#### 3.3.2. Blood Test Results

Among secondary outcome measures, the blood test results were analyzed for ESR (mm/h) (n = 98), CRP (mg/L) (n = 156), and RF (U/mL) (n = 156) using data from Li 2015 [40] and Wang 2016(1) [43]. The pooled results showed significant differences between the two groups regarding ESR, CRP, and RF (ESR: MD = 24.46, 95% CI: 20.72–28.20, *p* < 0.00001; CRP: MD = 3.11, 95% CI: 0.66–5.57, *p* = 0.01; RF: MD = 75.35, 95% CI: 57.02–93.67, *p* < 0.00001). The heterogeneity was 29% for CRP, 88% for RF, and not calculable for ESR (Figure 3).

#### 3.3.3. Other Outcome Measures of RA

Two studies (n = 88) [43,44] reported DAS28 data from baseline to endpoint, showing a significant difference (MD = 2.86, 95% CI: 2.56–3.16, *p* < 0.00001) with 0% heterogeneity (Figure 4).

### 3.4. Efficacy Assessment of WTD Combination Therapy

#### 3.4.1. Effective Rate

Among 14 studies, 13 studies (n = 1248) reported ER results [45,46,48,49,50,51,52,53,54,55,56,57,58]. Of these, refs. [45,46,48,49,50,51,53,54,55] ER was classified into four scales in nine studies, three scales were used in two studies [52,57], and two scales were used in two [56,58]. The significant differences varied based on the number of ER scales; however, significant results were obtained when all 13 studies were analyzed (RR = 1.25, 95% CI: 1.18–1.33, *p* < 0.00001) with 0% heterogeneity (Figure 5).

#### 3.4.2. Disease Activity Outcomes

Regarding TJC, the combined results of five studies (n = 513) [45,46,47,50,55] showed significant differences between the two groups (MD = 2.29, 95% CI:2.14–2.45, *p* < 0.00001, I^2^ = 96%). Additionally, data on MS were gathered from four studies (n = 438) [45,46,50,56], demonstrating significant differences between the two groups (MD = 17.98, 95% CI:14.49–21.47, *p* < 0.00001, I^2^ = 77%) (Figure 6).

#### 3.4.3. Blood Test Results

Nine studies (n = 1049) [45,46,47,49,50,54,55,56,58] provided data on ESR (mm/h), eight studies reported on CRP levels (mg/L) (n = 993) [45,46,47,49,50,55,56,58], and five studies reported on RF levels (U/mL) (n = 514) [45,46,47,54,56]. The combined results showed significant differences in all three parameters (ESR: MD = 9.66, 95% CI: 8.88–10.43, *p* < 0.00001; CRP: MD = 6.25, 95% CI: 5.75–6.74, *p* < 0.00001; RF: MD = 4.90, 95% CI: 3.67–6.14, *p* < 0.00001). Regarding heterogeneity, ESR, CRP, and RF demonstrated high levels of heterogeneity (ESR: 88%; CRP: 92%, RF: 94%) (Figure 7). 

#### 3.4.4. Other Outcome Measures of RA

Four studies (n = 582) [45,46,47,58] provided data on DAS28 from baseline to endpoint, revealing a significant difference (MD = 0.82, 95% CI: 0.70–0.95, *p* < 0.00001). However, the I^2^ statistic indicated high heterogeneity (I^2^ = 92%). Additionally, three studies reported data on the ACR series (n = 324) [46,47,56] and HAQ scores (n = 448) [46,47,58], with pooled data also demonstrating significant differences (ACR20: MD = 1.27, 95% CI: 1.15–1.42, *p* < 0.00001; ACR50: MD = 1.52, 95% CI: 1.22–1.88, *p* = 0.0002; ACR70: MD = 1.48, 95% CI: 1.08–2.01, *p* = 0.01, HAQ: MD = 0.25, 95% CI: 0.17–0.34, *p* < 0.00001). Heterogeneity among the ACR series and HAQ studies was low, with I^2^ values of 0% (Figure 8).

### 3.5. Safety Assessment

Among the 19 studies included in the analysis, 11 studies [40,41,42,47,48,49,53,54,55,56] reported adverse events associated with WTD treatment. Of these, two studies [41,54] reported no severe adverse events, while the remaining five studies identified various adverse events: skin irritation (6/325 [0.02%] vs. 13/305 [0.04%]) [42,45,47,49,56], gastrointestinal problems, including nausea with vomiting (9/277 [0.03%] vs. 18/252 [0.07%]) [42,48,49,53,55,56], diarrhea with vomiting (8/116 [0.07%] vs. 11/116 [0.09%] [40,45], liver failure (2/67 [0.03%] vs. 3/67 [0.04%] [45], and decreased white blood cell count (0/70 [0.00%] vs. 1/50 [0.02%] [42]. One study reported three cases of nausea, vomiting, and diarrhea in the experimental group, while two cases of upper abdominal pain and acid reflux were identified in the control group [54] (Table 2).

### 3.6. Risk of Bias Assessment

Regarding random sequence generation, 10 studies demonstrated an unclear risk of selection bias without specifying the method of randomization. Five studies demonstrated a high risk of bias, while four studies showed a low risk of bias utilizing the visiting order method. Apart from random sequence generation, all 19 studies showed comparable results. However, owing to the lack of clear criteria for assessing performance and detection biases in each study, the risk of performance and detection bias remained unclear. The risk of attrition bias was low across all studies. Additionally, reporting bias was deemed low in all the studies, while the risk of other biases remained unclear across all included studies (Figure 9).

### 3.7. Sensitivity Analysis

We performed sensitivity analyses for ESR, CRP, and ER. We examined how efficacy figures (MD and RR) and heterogeneity changed when each trial was excluded individually. In monotherapy (five included studies), efficacy measures and heterogeneity changed significantly when excluding three studies (Liu [41] and Wei [42] for ER; and Wang [43] for CRP). In contrast, in combination treatment (14 included studies), two studies (Hu [55] for ESR and Zhou [56] for CRP) showed significant changes (Appendix A).

### 3.8. Publication Bias Assessment

The ER of combination therapy was assessed based on data from > 10 studies, promoting an evaluation of publication bias. Across the 13 studies included in the analysis, a generally symmetrical figure was observed, indicating no obvious publication bias (Figure 10).

### 3.9. Evidence Evaluation

Table 3 shows the summarized overall results of the GRADE evaluation. Subjective outcomes such as ER, TJC, and MS were downgraded by one level in monotherapy and combination therapy, owing to the risk of bias. Objective outcomes with < 200 participants, such as ESR, CRP, and RF in WTD monotherapy, were downgraded by two levels owing to serious imprecision. Considering the heterogeneity, ER and RF in monotherapy were downgraded by one level. For combination therapy, ESR, CRP, RF, TJC, and MS were also downgraded by one level. Finally, the efficacy of combination therapy was rated as “moderate” for ER, ESR, CRP, and RF. For monotherapy, ER, ESR, and CRP were rated as “low”, as were RF, ER, TJC, and MS for combination therapy. Lastly, the RF of monotherapy was rated as “very low” (Table 3).

## 4. Discussion

RA is a prevalent autoimmune systemic disease [64] characterized by joint deformities and functional impairments, which significantly affects patients with RA [65]. While DMARDs are established treatments that lower CRP and ESR [66,67], a growing interest exists in alternative rheumatic treatments, including natural ingredients, particularly those used in TCM, such as WTD [68]. Ba et al. demonstrate that WTD may suppress RA through various chemical mechanisms, employing safer and more patient-friendly approaches [23]. In contrast to previous SRs that explored various TCM formulations [69], our review focuses specifically on WTD, covering 19 studies with 1794 participants.

Among these 19 studies, those employing combination therapy (14 studies) constitute a larger proportion than those utilizing monotherapy (five studies). Chae et al. assessed the efficacy of Simiao Xiaobi decoction for RA through an SR [70]. In contrast to his study, which emphasizes combination therapy, Simiao Xiaobi decoction was mainly administered as monotherapy, and it demonstrated improvements in RA symptoms. Therefore, we conclude that either combination therapy or monotherapy can be appropriately administered based on the condition of the patient or preference, emphasizing that rational decision making in prescribing medication is essential during treatment.

The experimental group initially showed superior ER than those of the control group, with reductions observed in TJC and MS in monotherapy and combination therapy. Additionally, significant differences between WTD and the control group were observed in DAS28, ACR20, and 50 results. DAS28 results were reported in six studies [43,44,45,46,47,58], followed by assessments of HAQ improvement [46,47,48], Lansbury score, activities of daily living (ADL), and quality of life (QOL) [54]. Objective numerical indicators such as ESR, CRP, and RF were included as secondary outcome measures. In monotherapy and combination therapy, the experimental group exhibited significant differences in ESR and CRP than those of the control group. 

Significant differences in RF were observed between the two groups in monotherapy and combination therapies. However, data from monotherapy studies indicate a higher risk owing to significant heterogeneity between the two included studies. RF, a recognized diagnostic marker for RA [71], is elevated in over 70% of RA cases compared to < 15% in other forms of arthritis [72]. However, elevated RF levels are also common in other autoimmune conditions, such as systemic lupus and Sjogren’s disease [73]. Additionally, WTD demonstrated efficacy across several inflammatory diseases beyond RA, emphasizing the need to clarify the correlation between RA and RF. 

In the control groups of the included studies, MTX and LEF were the most common treatments, followed by NSAIDs. Moreover, among the nine studies with single-treatment controls, LEF was used in four. MTX is recognized as more effective than other csDMARDs [74], and LEF is often considered the primary treatment option for patients who cannot tolerate MTX [16]. Compared with the primary conventional synthetic DMARDs (csDMARDs) typically used in early stages, the favorable outcomes linked to WTD, with no reported side effects, are significant. This suggests that WTD may be a safe and effective option for initial RA treatment.

Regarding adverse events, skin irritation, gastrointestinal issues, including nausea with vomiting, liver failure, diarrhea with vomiting, and decreased WBC were reported. However, no significant differences were observed between the experimental and control groups, with the incidence rates of all adverse events lower in the experimental groups than in the control groups. For example, the combination of WTD and MTX results in fewer adverse events, including abnormal liver and renal function, than in the MTX group alone [20]. MTX is the most popular DMARD; however, MTX frequently induces gastrointestinal side effects such as nausea, vomiting, diarrhea, hepatotoxicity, pulmonary toxicity, and hematologic toxicity, which can lead patients to discontinue its use [75]. Therefore, the use of WTD in RA treatment warrants careful consideration.

These statistical results showed a promising potential for WTD. However, marked discrepancies exist between the study results and clinical practice. First, regarding the duration of DMARD treatment, medications such as MTX and LEF usually require 3–6 months to show noticeable improvement. However, eight of the selected studies reported significant effects compared to those of MTX or LEF within 12 weeks. [76]. Second, the recent clinical practice guidelines for RA emphasize medication strategies, guiding physicians to prescribe alternative medications if initial medications are ineffective. [77] In essence, RA treatment typically builds on previous therapeutic strategies [16,17,18], a factor not addressed in the selected studies. Therefore, applying these findings comprehensively in clinical practice may be challenging. Finally, while the included studies aim to compare the effectiveness of WTD with commonly used conventional medications in clinical practice, several limitations affect their practical application.

Another limitation is the low quality of the included studies. First, a high risk of bias errors exists owing to inaccurate random assignment and inadequate blinding procedures. Second, despite the efforts of the authors, gaps in comprehensive database coverage may exist. The number of studies on WTD monotherapy and combination therapies is limited, and some studies have short treatment durations, making it challenging to assess the clear efficacy of WTD in either monotherapy or combination therapy. Third, all studies were conducted in China, which may limit their generalizability to the global population. Most did not adhere to international journal standards, and some may have employed unrealistic statistical methods, potentially leading to inaccurate data or methodology. These discrepancies could affect the applicability of the findings to actual clinical practice. Fourth, while the indicators generally used the same units, the risk of numerical errors could exist owing to studies employing units that differ from the commonly used ones. As a result, this study contains some results with high heterogeneity (Figure 7). Fifth, the presence of studies with sex ratios differing from actual clinical practice [41,48] or identified in sensitivity analyses [40,41,42,43,55,56] undermines the credibility of the statistical results. Finally, the WTD used in the experimental group lacks consistency across different studies owing to varying additions and subtractions. Moreover, determining the optimal treatment is challenging because of the diverse types and durations of interventions employed in the control group.

In conclusion, this SR numerically demonstrated that WTD enhanced various indicators compared to traditional treatments, whether used as monotherapy or in combination therapy. Despite these positive statistical results, significant gaps in clinical practice make practical application challenging. Therefore, future studies should prioritize factors such as randomization assignment, blinding of participants or results, and outcome reviews. Additionally, as discussed by an experienced rheumatologist, clinical studies should reflect situations that could benefit patients with RA to reach a definitive conclusion.

## 5. Conclusions

This review highlights the therapeutic potential of WTD compared to conventional treatment based on statistical aspects. However, owing to the significant discrepancies with clinical practice and the low quality of the included studies, applying these findings in real-world clinical settings is challenging for physicians. Therefore, further studies with well-designed and larger patient cohorts are needed to draw definitive conclusions.

## Figures and Tables

**Figure 1 healthcare-12-01739-f001:**
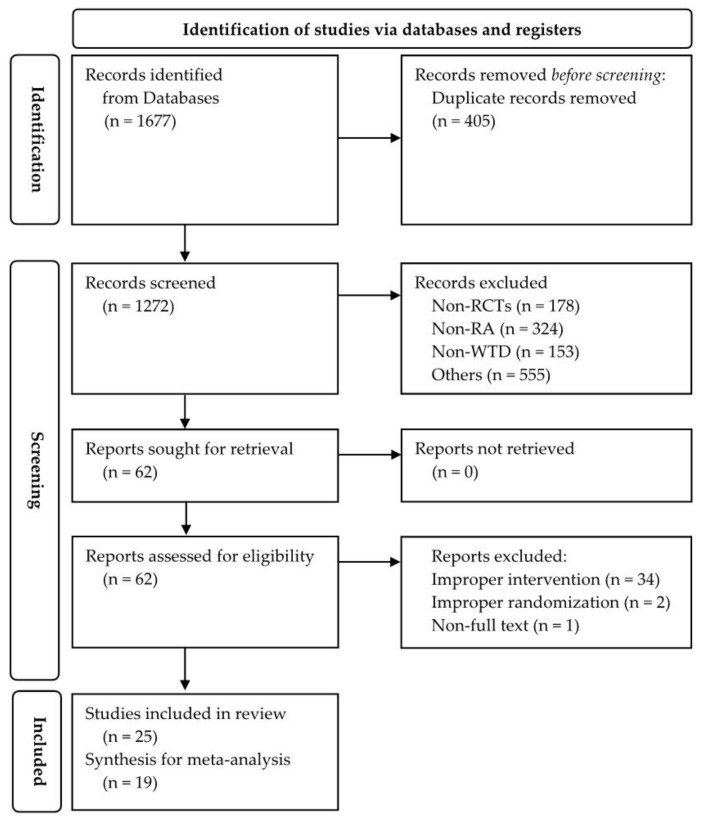
PRISMA flow diagram.

**Figure 2 healthcare-12-01739-f002:**
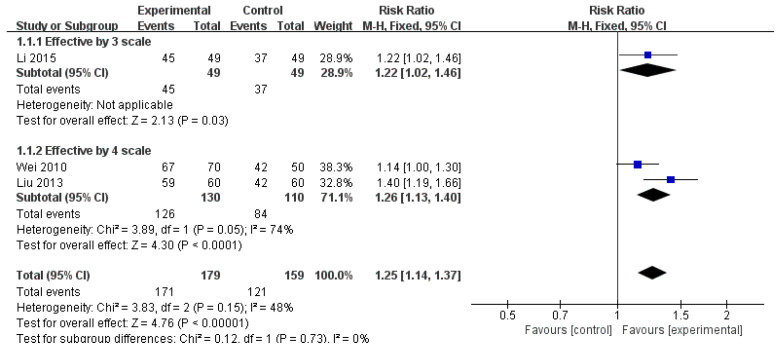
The comparison of WTD monotherapy vs. conventional treatment in ER. Abbreviations: CI, confidence interval [40,41,42].

**Figure 3 healthcare-12-01739-f003:**
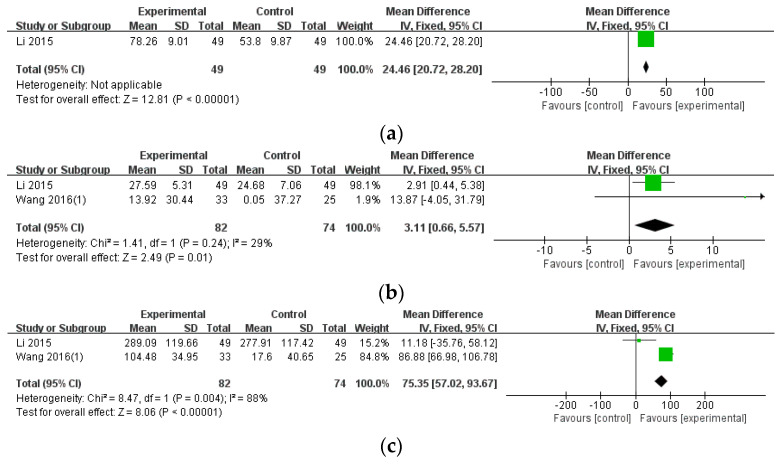
The comparison of WTD monotherapy vs. conventional treatment in blood test results. (**a**) Erythrocyte sedimentation rate (ESR) [40]; (**b**) C-reactive protein (CRP) [40,43]; (**c**) Rheumatoid factor (RF). Abbreviations: CI, confidence interval; SD, standard deviation [40,43].

**Figure 4 healthcare-12-01739-f004:**
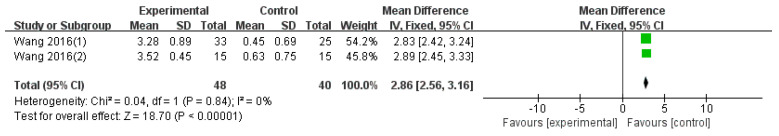
The comparison of WTD monotherapy vs. conventional treatment in DAS28. Abbreviations: CI, confidence interval; SD, standard deviation [43,44].

**Figure 5 healthcare-12-01739-f005:**
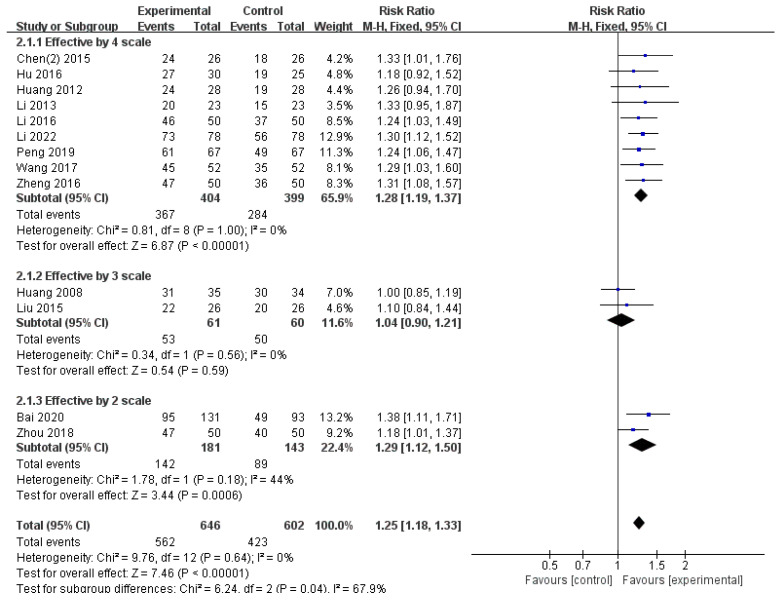
The comparison of WTD combination therapy vs. conventional treatment in ER. Abbreviations: CI, confidence interval [45,46,48,49,50,51,52,53,54,55,56,57,58].

**Figure 6 healthcare-12-01739-f006:**
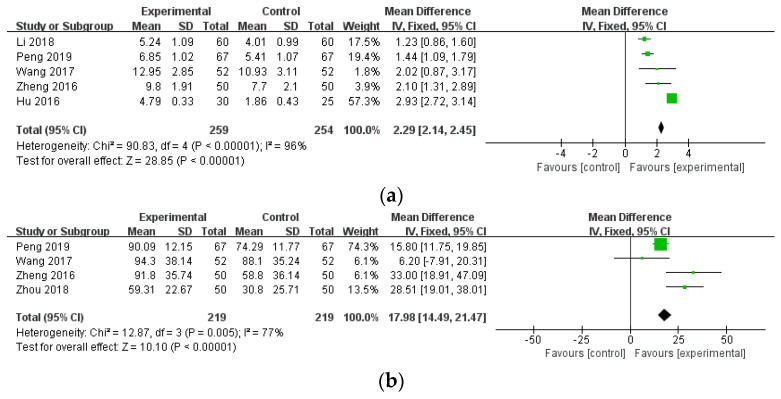
The comparison of WTD combination therapy vs. conventional treatment in disease activity outcomes. (**a**) Tender joint count (TJC) [45,46,47,50,55]; (**b**) Morning stiffness (MS) [45,46,50,56]. Abbreviations: CI, confidence interval; SD, standard deviation.

**Figure 7 healthcare-12-01739-f007:**
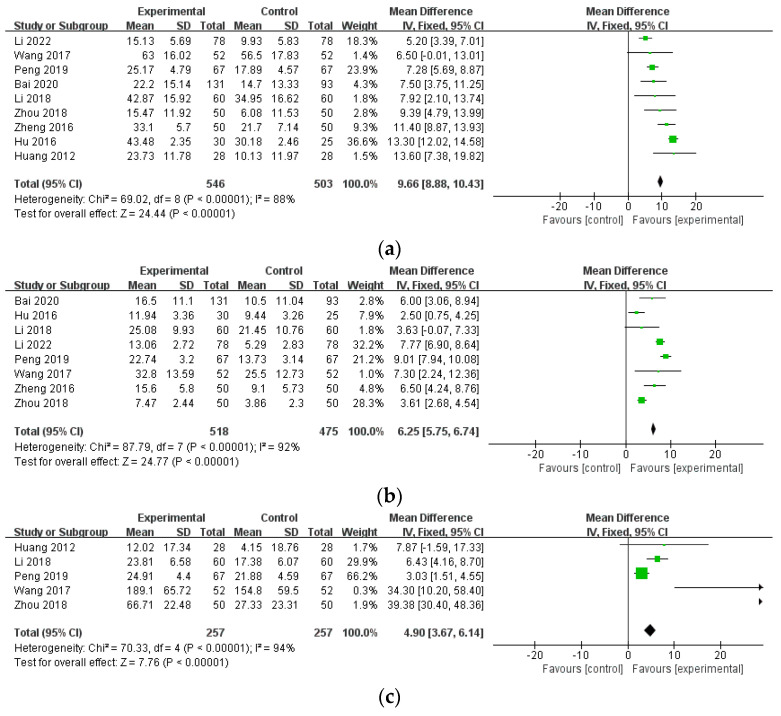
The comparison of WTD combination therapy vs. conventional treatment in blood test results. (**a**) Erythrocyte sedimentation rate (ESR) [45,46,47,49,50,54,55,56,58]; (**b**) C-reactive protein (CRP) [45,46,47,49,50,55,56,58]; (**c**) rheumatoid factor (RF) [45,46,47,54,56]. Abbreviations: CI, confidence interval; SD, standard deviation.

**Figure 8 healthcare-12-01739-f008:**
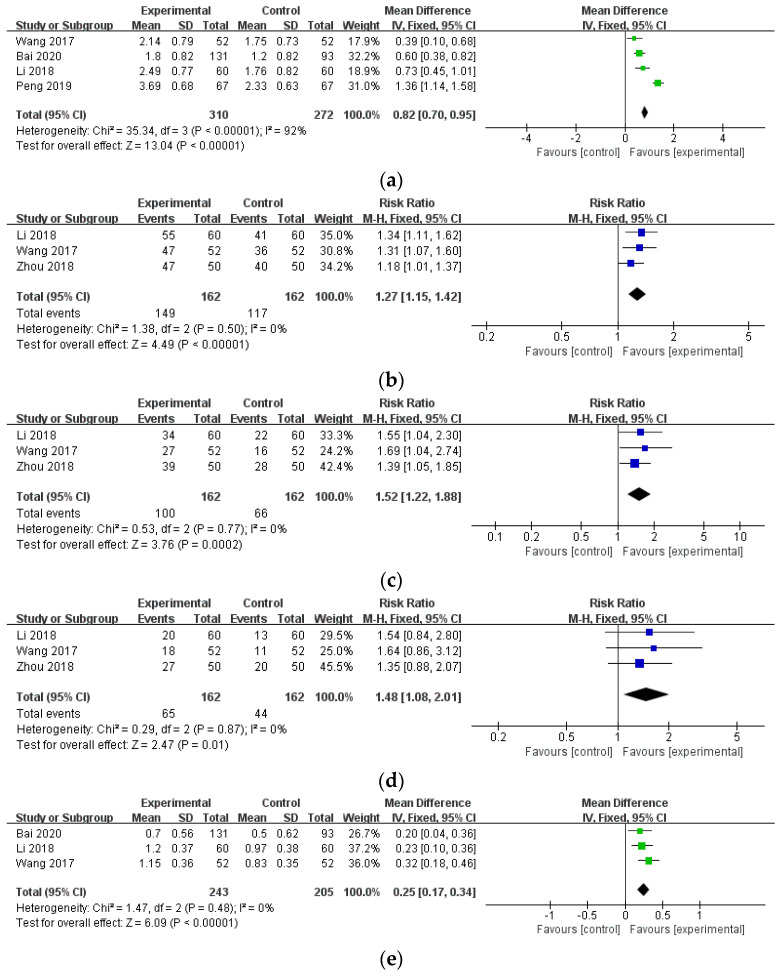
The comparison of WTD combination therapy vs. conventional treatment in other outcomes. (**a**) Disease activity score in 28 joints (DAS28)) [45,46,47,58]; (**b**) American College of Rheumatology 20 (ACR20) [46,47,56]; (**c**) ACR 50 [46,47,56], (**d**) ACR 70 [46,47,56], (**e**) Health assessment questionnaire (HAQ) [46,47,58]. Abbreviations: CI, confidence interval; SD, standard deviation.

**Figure 9 healthcare-12-01739-f009:**
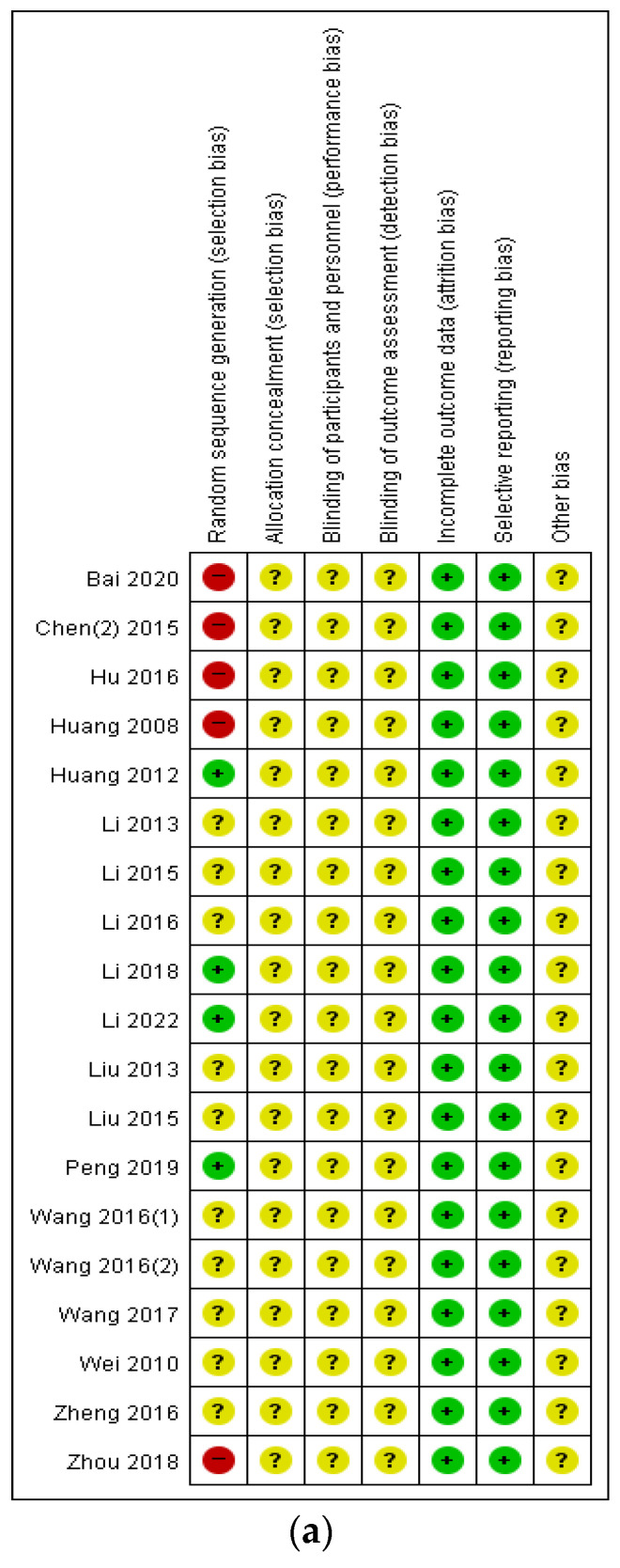
Risk of bias summary and graph. (**a**) summary [40,41,42,43,44,45,46,47,48,49,50,51,52,53,54,55,56,57,58]; (**b**) graph.

**Figure 10 healthcare-12-01739-f010:**
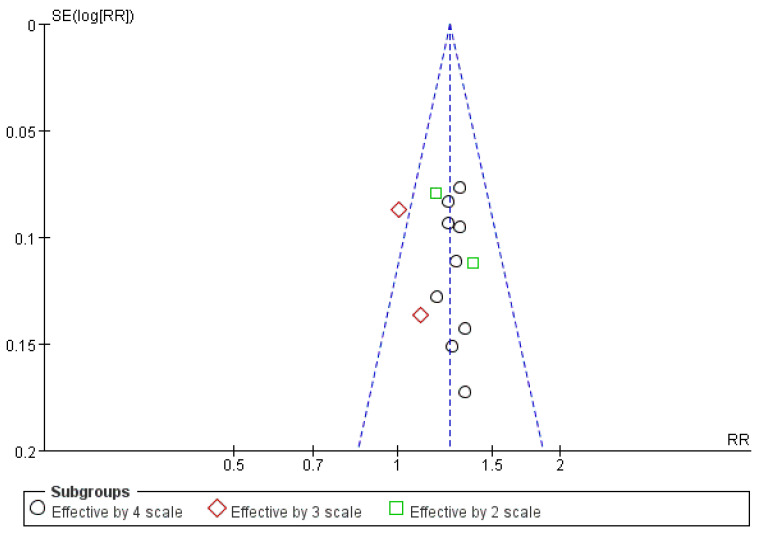
Funnel plot in ER.

**Table 1 healthcare-12-01739-t001:** The characteristics of the included studies.

Study (y)	Number of RA Patients (Male/Female)		Duration of RA		Treatment Details of WTD Group		Treatment Details of CTR Group		Duration of Treatment	Outcome Measures (Primary/Secondary/ Other Measures Used in Meta-Analysis)
	WTD	CTR	WTD	CTR	Name of Treatment	Frequency Sessions of Treatment	Name of Treatment Type of Administration (Oral or Injection)	Frequency Sessions of Treatment		
Li 2015 [40]	49 (26/23)	49 (24/25)	2.23 ±1.34 (m)	2.34 ±1.26 (m)	Modified WTD	BID	LEF (50 mg for first 3 d + 10~20 mg) by oral	QD	30 d	ER/ESR, CRP, RF, AEs
Liu 2013 [41]	60 (35/25)	60 (40/20)	1 ± 2.1 (y)	1 ±2.2 (y)	Modified WTD	QD	LEF (50 mg for first 3 d + 10 mg) by oral	QD	30 d	ER/AEs
Wei 2010 [42]	70 (30/40)	50 (24/26)	2–8 (y)	1–7(y)	Modified WTD	BID	MTX (15 mg), Prednisone (10 mg), Nimesulide (0.1 g), Glucosides capsule (2 cap) by oral	BID, TID, BID, TID	4 w	ER/AEs
Wang 2016(1) [43]	33 (16/17)	25 (12/13)	11.94 ±8.02 (y)	8.04 ±6.98 (y)	WTD	BID	MTX (15 mg) by oral	QW	8 w	CRP, RF/DAS28/ Cyclic citrullinated peptide antibody
Wang 2016(2) [44]	15 (8/7)	15 (7/8)	7.67 ±3.12 (y)	7.27 ±3.69 (y)	WTD	BID	MTX (15 mg) by oral	QW	8 w	DAS28/TNF-a, IL-6, Vascular endothelial growth factor, IL-17, HGB, PLT
Peng 2019 [45]	67 (20/47)	67 (19/48)	7.28 ±3.65 (m)	7.33 ±3.54 (m)	Modified WTD +CTR	BID	LEF (20 mg), MTX (10 mg) by oral	QD, QW	3 m	ER, TJC, MS/ESR, CRP, RF, AEs/ DAS28, SJP
Wang 2017 [46]	52 (26/26)	52 (28/24)	4.4 ±2.1 (NR)	4.7 ±2.2 (NR)	Modified WTD +CTR	QD	LEF (20 mg), MTX (10 mg) by oral	QD, QW	3 m	ER, TJC, MS/ESR, CRP, RF/ DAS28, ACR20, ACR50, ACR70, HAQ, SJP
Li 2018 [47]	60 (24/36)	60 (26/34)	4.49 ±3.75 (y)	4.37 ±3.42 (y)	Modified WTD + CTR	TID	Meloxicam Tab (7.5 mg), DCF (NR), Baishao GLS Cap (2 Cap) by oral	QD, TID, TID	4 w	TJC/ESR, CRP, RF, AEs/ DAS28, ACR20, ACR50, ACR70, HAQ, SJP
Chen 2015(2) [48]	26 (19/7)	26 (21/5)	6.87 ±2.33 (y)	7.57 ±1.98 (y)	Modified WTD +CTR	QD	MTX (10 mg), Tripterygium GLS (20 mg) by oral	QW	6 m	ER/AEs
Li 2022 [49]	78 (33/45)	78 (32/46)	2.23 ±1.34 (m)	2.34 ±1.26 (m)	Modified WTD +CTR	BID	DCF (25 mg), LEF (10 mg) by oral	TID, QD	30 d	ER/ESR, CRP, AEs
Zheng 2016 [50]	50 (23/27)	50 (22/28)	6.5 ±5.4 (y)	5.8 ±6.3 (y)	WTD +CTR	BID	LEF (20 mg) by oral	BID	6 m	ER, TJC, MS/ESR, CRP
Li 2016 [51]	50 (29/21)	50 (NR/NR)	6.88 ±2.34 (y)	7.56 ±1.97 (y)	Modified WTD +CTR	BID	MTX (10 mg), Tripterygium GLS (20 mg) by oral	QW, TID	3 m	ER
Liu 2015 [52]	26 (NR/NR)	26 (NR/NR)	NR	NR	Modified WTD +CTR	TID	LEF (20 mg) by oral	QD	3 m	ER
Li 2013 [53]	23 (NR/NR)	23 (NR/NR)	NR	NR	Modified WTD +CTR	BID	DCF (0.1 g), SSZ (0.75 g), Tripteryzgium GLS (0.02 g) by oral	QD, TID, TID	1 m	ER/AEs
Huang 2012 [54]	28 (NR/NR)	28 (NR/NR)	NR	NR	Modified WTD +CTR	BID	^99m^Tc-Methyl diphosphonate (5 mg) by injection	NR	2 m	ER/ESR, RF, AEs
Hu 2016 [55]	30 (NR/NR)	25 (NR/NR)	NR	NR	WTD +CTR	NR	DCF (50 mg), MTX (10 mg), LEF (20 mg) by oral	BID, QW, QD	15 d	ER, TJC/ESR, CRP, AEs
Zhou 2018 [56]	50 (32/18)	50 (28/22)	39.20 ±11.38 (m)	39.12 ±11.04 (m)	Modified WTD +CTR	BID	Igurtimod (25 mg) by oral	BID	3 m	ER, MS/ESR, CRP, RF, AEs/ ACR 20, ACR 50, ACR70/ADL, QOL, SJP, Joint pain, Joint swelling, Joint heatness, Joint hardness/Lansbury Score
Huang 2008 [57]	35 (NR/NR)	34 (NR/NR)	NR	NR	Modified WTD +CTR	BID	Ibuprofen (0.3 mg), MTX (15 mg) by oral	BID, QW	6 w	ER
Bai 2020 [58]	131 (29/102)	93 (17/76)	1.0 ±0.5 (y)	0.9 ±0.4 (y)	Modified WTD +CTR	QD	MTX (7.5–15 mg) by oral	QW	12 w	ER/ESR, CRP/DAS28, HAQ/LDA
Luo 2008 [59]	36 (NR/NR)	34 (NR/NR)	NR	NR	Modified WTD +CTR	BID	Celecoxib (0.2 g), MTX (15 mg)	BID, QW	6 w	TJC, MS/ESR, CRP, RF, AEs/ GS, WT20, BPC
Luo 2009 [60]	32 (NR/NR)	30 (NR/NR)	NR	NR	Modified WTD +CTR	BID	Celecoxib (0.2 g), MTX (15 mg)	BID, QW	6 w	TJC, MS/ESR, CRP, RF, AEs/ GS, WT20
Mao 2013 [61]	32 (NR/NR)	30 (NR/NR)	NR	NR	Modified WTD +CTR	BID	Ibuprofen (0.3 g), MTX (15 mg)	BID, QW	6 w	TJC, MS/ESR, CRP, RF, AEs/ SJP, GS, WT20, RBC, HGB
Zheng 2008 [62]	32 (NR/NR)	30 (NR/NR)	NR	NR	Modified WTD +CTR	BID	Celecoxib (0.2 g), MTX (15 mg)	BID, QW	6 w	TJC, MS/ESR, CRP, RF, AEs/ GS, WT20, PLT
Zheng 2012 [63]	32 (NR/NR)	30 (NR/NR)	NR	NR	Modified WTD +CTR	BID	Ibuprofen (0.3 g), MTX (15 mg)	BID, QW	6 w	TJC, MS/ESR, CRP, RF, AEs/Grip strength, 20-min walking time, TNF-a, IL-6, PAF
Yue 2019 [34]	40 (NR/NR)	37 (NR/NR)	NR	NR	WTD +CTR	NR	MTX (10–15 mg)	QW	12 w	CRP, ESR, AEs/DAS28, HAQ/LDA, Remission rate, Overall response rate

Abbreviations: ACR: American College of Rheumatology; ADL: Activities of daily living; AEs: Adverse events; BID: Bis in die (twice a day); BPC: Blood platelet count; Cap: Capsule; CRP: C-reactive protein; CTR: Control; d: day; DAS28: Disease activity score 28; DCF: Diclofenac; ER: Effective rate; ESR: Erythrocyte sedimentation rate; GLS: Glycosides; GS: Grib strength; HAQ: Health Assessment questionnaire; HGB: Hemoglobin; IL: Interleukin; LDA: Low disease activity; LEF: Leflunomide; m: month; MS: Morning stiffness; MTX: Methotrexate; NR: Not reported; PAF: Platelet activating factor; PLT: Platelet; QD: Quique die (once daily); QOL: Quality of life; QW: Once a week; RA: Rheumatoid arthritis; RBC: Red Blood Cell; RF: Rheumatoid factor; SJP: Swollen joint point; SSZ: Sulfasalazine; TID: Ter in die (three times a day); TJC: Tender joint count; w: week; WTD: Wu tou decoction; WT20: Walking time for 20 m; y: year.

**Table 2 healthcare-12-01739-t002:** Adverse events in each group.

Adverse Events	Total Number of Adverse Events	
	Treatment Group	Control Group
Skin irritation	6 [42,45,47,49,56]	13 [42,45,47,49,56]
Gastrointestinal problem (with nausea and vomiting)	9 [42,48,49,53,55,56]	18 [42,48,49,53,55,56]
Diarrhea with vomiting	8 [40,45]	11 [40,45]
Liver failure	2 [45]	3 [45]
Decreased WBC	0 [42]	1 [42]
Upper abdominal pain and reflux	0 [54]	2 [54]

**Table 3 healthcare-12-01739-t003:** GRADE certainty of evidence assessments.

Outcome	Included RCTs (Participants)	Effect Estimate (95% CI)	I^2^	Quality of Evidence	Reasons
ER (monotherapy)	3 (338)	RR 1.25 (1.14, 1.37)	48%	⨁⨁◯◯ Low	Risk of bias, inconsistency
ESR (monotherapy)	1 (98)	MD 24.46 (20.72, 28.20)	Not applicable	⨁⨁◯◯ Low	Serious imprecision
CRP (monotherapy)	2 (156)	MD 3.11 (0.66, 5.57)	29%	⨁⨁◯◯ Low	Serious imprecision
RF (monotherapy)	2 (156)	MD 75.35 (57.02, 93.67)	88%	⨁◯◯◯ Very Low	Serious imprecision, inconsistency
ER (combination therapy	13 (1248)	RR 1.25 (1.18, 1.33)	0%	⨁⨁⨁◯ Moderate	Risk of bias
TJC (combination therapy)	5 (513)	MD 2.29 (2.14, 2.45)	96%	⨁⨁◯◯ Low	Risk of bias, inconsistency
MS (combination therapy)	4 (438)	MD 17.98 (14.49, 21.47)	77%	⨁⨁◯◯ Low	Risk of bias, inconsistency
ESR (combination therapy)	9 (1049)	MD 9.66 (8.88, 10.43)	88%	⨁⨁⨁◯ Moderate	inconsistency
CRP (combination therapy)	8 (993)	MD 6.25 (5.75, 6.74)	92%	⨁⨁⨁◯ Moderate	inconsistency
RF (combination therapy)	5 (514)	MD 4.90 (3.67, 6.14)	94%	⨁⨁⨁◯ Moderate	inconsistency

GRADE has four levels of quality of evidence: high, moderate, low, and very low. More ⨁ indicates better quality of evidence. Abbreviations: CRP: C-reactive protein; ER: Effective rate; ESR: Erythrocyte sedimentation rate; MD: Mean difference; MS: Morning stiffness; RF: Rheumatoid factor; RR: Risk ratio; TJC: Tender joint count.

## Data Availability

Data are available from the corresponding author on reasonable request.

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
