# Peer review of "The Effectiveness and Safety of Wu Tou Decoction on Rheumatoid Arthritis—A Systematic Review and Meta-Analysis"

_healthcare, 2024, doi:10.3390/healthcare12171739_

Round 1

Reviewer 1 Report (Previous Reviewer 1)

Comments and Suggestions for Authors

Dear authors,

The meta-analysis of studies with WTD in RA is being resubmitted after adding a few passages of text, after it had been rejected due to fundamental concerns. The concerns relate primarily to the underlying studies, which are not suitable for assessing the effectiveness and tolerability of long-term treatment of rheumatoid arthritis due to their brevity and the endpoints collected.

The authors are convinced of the effect of WTD. They state this in the abstract and introduction: "its significant anti-inflammatory properties". Despite mentioning the methodological limitations, the authors stick to the conclusion in their overall assessment and especially in their abstract: "WTD demonstrated better improvement than conventional treatments".

It is astonishing how the authors can use the ACR/EULAR criteria for patient selection in studies that were published before the ACR/EULAR criteria were published.

Since the data are interesting and important and there is probably a potential of the WTD in symptom control that can improve the time until DMARD therapy takes effect for the patient, the results should be put into the context of the disease and re-sorted and discussed by an experienced rheumatologist.

Studies that are of very questionable quality/recruitment such as Chen 2015 with 40 male and 12 female participants, Liu 2013 (75m/45f) and Zhou 2018 (60/40) in a disease that affects 3/4 female patients should not be evaluated.

Author Response

Thank you for your valuable review. We checked various points you raised and responded the questions in a sincere manner. We wrote our answer in blue, manuscript text in red, and modified sentence in line. Additionally, we have once again proofread the parts that we have revised (e.g., abstract, introduction, discussion, and conclusion section). We attach the MS word file, please see it.

Reviewer 2 Report (Previous Reviewer 2)

Comments and Suggestions for Authors

Thank you for the revision. The manuscript has improved substantially compared to the previous version. The methods for meta-analysis is well described and the discussion is balanced. Though I am skeptical of the practical merits of the WTD, the paper may be of interest to the general readerships of the journal.

Author Response

Thank you for your valuable review. We checked various points you raised and responded the questions in a sincere manner. We wrote our answer in blue, manuscript text in red, and modified sentence in line. Additionally, we have once again proofread the parts that we have revised (e.g., abstract, introduction, discussion, and conclusion section). We attach the MS word file, please see it.

Reviewer 3 Report (New Reviewer)

Comments and Suggestions for Authors

Dear Author(s),

I would like to extend my congratulations on your recent publication addressing the effectiveness and safety of Wu Tou decoction in the treatment of rheumatoid arthritis through a systematic review and meta-analysis. I have several points for clarification and suggestions for improvement, as outlined below:

  • The abstract is challenging to comprehend due to the excessive use of abbreviations, many of which are unnecessary as they are not utilized frequently within the abstract.

  • The introduction lacks depth and breadth. A more comprehensive introduction is necessary, thoroughly analyzing the existing literature on this topic. Additionally, a clear justification for the need for this systematic review and meta-analysis should be provided.

  • The search strategy section does not include the specific terms employed in the search, which is essential for reproducibility and transparency.

  • The discussion contains several confusing paragraphs that do not directly address the article's objectives, making it difficult to follow. These sections should be rewritten for clarity and coherence.

  • The conclusion should be presented without the use of abbreviations to ensure clarity and ease of understanding.

  • It is recommended to incorporate more recent bibliographic references, particularly systematic reviews published within the last three years, to ensure the study's relevance and timeliness.

Author Response

Thank you for your valuable review. We checked various points you raised and responded the questions in a sincere manner. We wrote our answer in blue, manuscript text in red, and modified sentence in line. Additionally, we have once again proofread the parts that we have revised (e.g., abstract, introduction, discussion, and conclusion section). We attach the MS word file, please see it.

Round 2

Reviewer 1 Report (Previous Reviewer 1)

Comments and Suggestions for Authors

Dear authors,

thank you for the revision and the comments.

You write "ACR was first published in 1987 and is still being updated (EULAR was first published in 2010)" (and cite the 2010 criteria in [23]). "And our included studies were published between 2010 and 2020".

That is not true. The ACR (formerly ARA) criteria were first published in 1956, revised repeatedly and only revised in 1987. The 2010 criteria show a clear fundamental change compared to the previous ones and are ultimately not an "update". Study reference [47], for example, was published in 2009, i.e. before 2010. It is formally correct but scientifically incorrect to write "We included studies involving patients with RA using authoritative criteria such as the American College of Rheumatology (ACR) and European League Against Rheumatism (EULAR) [23]" in the inclusion criteria if you yourself do not know exactly which criteria were used. You should then state this in the results section, or omit this part.

In the "Introduction" you write "WTD is an herbal medicine that has been used for over a thousand years to treat RA." If it were true, that would be very interesting. The first description of rheumatoid arthritis accepted by scientific standards dates back to 1800 [Entezami P, Fox DA, Clapham PJ, Chung KC. Historical perspective on the etiology of rheumatoid arthritis. Hand Clin. 2011 Feb;27(1):1-10. doi: 10.1016/j.hcl.2010.09.006. PMID: 21176794; PMCID: PMC3119866.], so it is a good 200 years old, and the authors have not found any published paleopathological studies of rheumatoid arthritis from China either.

You further write that WTD inhibits NF-κB phosphorylation (through the action of Herba ephedrae) and enhances nuclear factor-like 2 (Nrf2) expression (via Radix Astragali and Radix aconiti) [15] without specifying that this study does not refer to RA but to rats (which are not a model for RA).

Author Response

Thank you for your valuable review. We checked various points you raised and responded the questions in a sincere manner. We wrote our answer in blue, manuscript text in red, and modified sentence in line.

Thank you again for your comments and suggestions, we could make our manuscript more appropriate and scientific.

Reviewer 3 Report (New Reviewer)

Comments and Suggestions for Authors

-

Author Response

Thank you for your valuable review. We checked various points you raised and responded the questions in a sincere manner. We wrote our answer in blue, manuscript text in red, and modified sentence in line.

Thank you again for your comments and suggestions, we could make our manuscript more appropriate and scientific.

This manuscript is a resubmission of an earlier submission. The following is a list of the peer review reports and author responses from that submission.

Round 1

Reviewer 1 Report

Comments and Suggestions for Authors

Dear authors, for a meta-analysis you searched for all published studies on WTD in various databases and then selected 19 studies for further evaluation. 5 studies tested WTD against basic therapy, 14 studies tested WTD + CTR against CTR. It is not clear from the statements whether the CTR was newly initiated or continued.

In their methodology they describe the effect of WTD against control, but in the latter 14 studies the effect of WTD against placebo was supposedly examined with basic therapy started in parallel or ongoing therapy (information on this is missing).

The text reveals an insufficient knowledge of rheumatoid arthritis (RA), its surrogates and principles by the authors of the meta-analysis and supposedly the studies cited.

RA is not an auto-inflammatory disease!

Studies with a duration of 15 days to 3 months cannot show an effect of a newly initiated DMARD therapy because this occurs later. Therefore they cannot serve as a comparator.

The unit of RF in g/L is completely unusual, the correct unit would be U/ml. A change in the RF in the short period of time is not to be expected due to the physiological half-life during the study period, and the reported improvements are therefore not credible; in fact, they raise doubts about the overall credibility.

"ER" is not a common progression parameter.

There are no RA diagnostic criteria, only classification criteria.

Contrary to what the authors write, injections are also medication.

The rate of reported side effects with conventional therapy is incredibly low and clearly discrepant to all controlled studies.

The fact that (significantly) fewer side effects occur with WTD + conventional therapy than with conventional (same) therapy is not credible and understandable and suggests methodological errors or incorrect reports.

The table uses abbreviations that are not explained (T-614, Tc-MDP).

The statement “This can suppress RA through various channels with various chemical effects accompanied with safer and human-friendly methods” is not supported by the studies.

The images are all of inadequate quality.

Unfortunately, the original publications used as sources cannot be viewed.

The authors mention some of the limitations of the studies.

Comments on the Quality of English Language

The wording could be improved, perhaps by re-translation or LL-modelling.

Author Response

Thank you for your valuable review. We checked various points you raised and responded the questions in a sincere manner. Please see the attachment. We wrote our answer in blue, manuscript text in red, and modified sentence in line. 

Regarding 19 included studies, we attempted to attach zip files, but were unable to do so due to system issues (documents must be in Word or PDF format). First, we will send it to the editorial office. We apologize for any inconvenience.

Reviewer 2 Report

Comments and Suggestions for Authors

The systematic review and meta-analysis is well written, with clear methods and quality assessment. WTD appeared to provide some benefit in addition to the standard therapy for rheumatoid arthritis, albeit with limited quality of evidence. I suggest adding some statement on where WTD may have a role in the clinical therapy for rheumatoid arthritis.

Comments on the Quality of English Language

no comment

Author Response

Thank you for your valuable review. We checked various points you raised and responded the questions in a sincere manner. Please see the attachment. We wrote our answer in blue, manuscript text in red, and modified sentence in line. 

Reviewer 3 Report

Comments and Suggestions for Authors

I appreciate the opportunity to review this manuscript.

The authors perform a systematic review and meta-analysis on the effect of WTD in rheumatoid arthritis.

Introduction. This section is essential for the reader to understand the object of study, as well as the importance of the study. It seems poorly structured, since it does not first address the pathology under study but starts by poorly describing the included therapy.

Lines 50-58 should explain in more detail the pathways modulating inflammation.

Lines 68-69, you cannot say in a scientific article that “it seems to be a reasonable alternative...”

The objectives of a systematic review cannot be to “minimize adverse events.” Systematic reviews aim to bring together the most recent or significant findings in an area of study but no such conclusions can be drawn without an empirical study.

Materials and methods. It is striking that the protocol was conducted in 2020, registered in 2022, and uploaded for publication in 2024. This suggests that relevant articles may have been published after the date of the systematic review.

Some of the databases used are questionable, and others as relevant as WoS, among others, are not included. In addition, it is strange that the search terms are in different languages depending on the database consulted.

The inclusion criteria are ambiguous and exclusion criteria are not included.

p's should be in italics and lowercase throughout the document.

The results are presented in different ways for each study and the way of analyzing the data in each of them is not explained.

In the discussion, the authors again summarize the treatment and results but do not discuss or compare with other studies. They are simply restated.

Author Response

(The authors gave the same response as above.)

Round 2

Reviewer 1 Report

Comments and Suggestions for Authors

Dear authors,

thank you for the original publications, revisions and comments.

Unfortunately, it is not clear in the script provided which parts of the text have been deleted, only the added ones are marked. This makes it difficult to read and make a final assessment.

I still wonder what is measured by swollen joint point (SJP), I have not found anything about it in the usual literature. You probably mean "count".

The ACR classification criteria are not diagnostic criteria, that has not been changed.

The data necessary for rheumatologists on the duration of the disease, previous therapies, gender, etc. are still missing - as is probably the case in the original publications.

When analyzing the data, a discrepancy between painful and swollen joints (if SJP actually means SJC) is noticeable in many cohorts, with averages of less than 2 affected joints in some cases. The number of painful joints is then sometimes ten times higher. At the same time, ESR and RF are very high. This may be typical for Chinese cohorts, but it is not the case in the usual cohorts. The high co-incidence of viral hepatitis could play a role here.

I also consider such a rapid and significant improvement in the rheumatoid factor reported in the studies to be immunologically implausible and a marker for dubious results.

The authors cite studies with JAK inhibitors as an argument for predictable rapid improvements, but the controls were MTX or leflunomide, both substances for which improvements are only expected after 3-6 months.

The transfer of the data on traditional healing methods, which are published entirely in China and predominantly in Chinese, continues to be a problem with scientific standards and is very dependent on the quality of the original publications. The publications presented do not reflect the standard expected for international journals.

I therefore continue to consider the strength of the conclusions from the meta-analysis to be unfounded.

Comments on the Quality of English Language

Unfortunately, it is not clear from the script provided which parts of the text have been deleted; only the added parts are marked. This makes it difficult to read and make a final assessment.

In addition, minor linguistic deficiencies are noticeable in the additions.

Reviewer 3 Report

Comments and Suggestions for Authors

The methodology used to conduct a systematic review is still insufficient. While the authors have used known databases, they have left articles unreviewed and not included that should appear in this manuscript for the information to be adequate and sufficient.

I encourage the authors to perform a new search, including international databases such as ScienceDirect, WoS, Wiley, etc., so that the manuscript may contain relevant information on the object of study.